# Genetic Mouse Models as In Vivo Tools for Cholangiocarcinoma Research

**DOI:** 10.3390/cancers11121868

**Published:** 2019-11-26

**Authors:** Oihane Erice, Adrian Vallejo, Mariano Ponz-Sarvise, Michael Saborowski, Arndt Vogel, Diego F. Calvisi, Anna Saborowski, Silvestre Vicent

**Affiliations:** 1Center for Applied Medical Research, Program in Solid Tumors, University of Navarra, 31008 Pamplona, Spain; oeazparren@unav.es (O.E.); avallejo.3@alumni.unav.es (A.V.); 2Department of Medical Oncology, Clinica Universidad de Navarra, 31008 Pamplona, Spain; mponz@unav.es; 3IdiSNA, Navarra Institute for Health Research, 31008 Pamplona, Spain; 4Department of Gastroenterology, Hepatology, and Endocrinology, Hannover Medical School, 30625 Hannover, Germany; saborowski.michael@mh-hannover.de (M.S.); vogel.arndt@mh-hannover.de (A.V.); 5Institute for Pathology, Regensburg University, 93053 Regensburg, Germany; Diego.Calvisi@klinik.uni-regensburg.de; 6Centro de Investigación Biomédica en Red de Cáncer (CIBERONC), 28029 Madrid, Spain

**Keywords:** cholangiocarcinoma, biliary tract, hydrodynamic injection, genetically-engineered mice, in vivo models

## Abstract

Cholangiocarcinoma (CCA) is a genetically and histologically complex disease with a highly dismal prognosis. A deeper understanding of the underlying cellular and molecular mechanisms of human CCA will increase our current knowledge of the disease and expedite the eventual development of novel therapeutic strategies for this fatal cancer. This endeavor is effectively supported by genetic mouse models, which serve as sophisticated tools to systematically investigate CCA pathobiology and treatment response. These in vivo models feature many of the genetic alterations found in humans, recapitulate multiple hallmarks of cholangiocarcinogenesis (encompassing cell transformation, preneoplastic lesions, established tumors and metastatic disease) and provide an ideal experimental setting to study the interplay between tumor cells and the surrounding stroma. This review is intended to serve as a compendium of CCA mouse models, including traditional transgenic models but also genetically flexible approaches based on either the direct introduction of DNA into liver cells or transplantation of pre-malignant cells, and is meant as a resource for CCA researchers to aid in the selection of the most appropriate in vivo model system.

## 1. Introduction

Cholangiocarcinoma (CCA) is the most prevalent biliary neoplasm and the second most frequent liver tumor after hepatocellular carcinoma (HCC), accounting for 10–20% of cancer-related deaths due to hepatobiliary malignancies [1,2]. CCA involves a heterogeneous group of cancers arising from the biliary tree that can be classified based on their anatomical location as intrahepatic (iCCA), perihiliar (pCCA) and distal (dCCA) [3,4]. Biliary inflammation is the common denominator of several risk factors identified thus far, such as primary sclerosing cholangitis, infestation with liver flukes, hepatitis C or B virus infection, chronic alcohol abuse, and non-alcoholic steatohepatitis. Most CCAs, however, are considered sporadic and arise in the absence of evident risk factors.

Surgical resection remains the only potentially curative therapeutic option, but only one-third of patients qualify for surgery at the time of diagnosis and early recurrence worsens the prognosis. Since the disease frequently remains clinically silent for a long time, most patients are bound to receive palliative treatments from initial diagnosis onwards. Based on the results from the ABC-02 trial, the standard chemotherapeutic regimen consists of gemcitabine in combination with cisplatin, leading to a median overall survival of 11.7 months with a median progression-free survival of 8 months [5]. Retrospective analyses regarding the use of second line treatments have been inconclusive and, thus far, no second line treatment recommendations exist, further emphasizing the imperative need for novel and more effective therapeutic strategies.

The molecular landscape of cholangiocarcinoma is heterogeneous and segregates with the anatomical location as well as the underlying risk factors. Based on the results from recent sequencing studies, an extensive catalogue of genetic alterations exists, including oncogenic activation of *KRAS*, or inactivating mutations in tumor suppressor genes (TSGs) such as *TP53*, *CDKN2A/B*, *SMAD4* and *PTEN*, as well as in DNA repair genes (e.g., *BRCA*) and epigenetic modifiers such as *ARID1A*. Notably, in about 30–40% of CCA patients, molecularly targetable alterations exist, indicating that precision medicine may be beneficial in a considerable subset of biliary tract cancers. Indeed, several of such “targeted” agents have already been tested within the framework of clinical trials, including FGFR inhibitors in patients with *FGFR2* fusions, IDH inhibitors in patients with *IDH1/2* mutations, and BRAF/MEK inhibitors in patients with activating *BRAF* mutations. Despite the increasing number of clinical trials, the early positive signals for precision medicine and an expanding “toolbox” for the treatment of CCA patients, we are still lacking a “deeper” understanding of those complex mechanisms that lead to the development of biliary cancer and determine the response or resistance to therapy.

As the genetic annotation of human cancer evolved, a plethora of genetically engineered murine models of cancer have been developed, which have since then served as pre-clinical platforms that allow us to study the disease in the context of a clinically relevant, intact microenvironment. With the increasing clinical and scientific recognition of biliary tract cancers, a repertoire of murine model systems for CCA has been developed in recent years and is now at our disposal to choose from. Considering the heterogeneity of the disease and the vast array of open questions regarding CCA pathophysiology, it is highly unlikely, though, that one single model will serve as “the ultimate”, universal pre-clinical tool. In this review, we will discuss a selection of murine models that have the potential to accelerate CCA research, expand our current knowledge about this cancer and, eventually, unveil novel opportunities to build better treatment strategies.

## 2. Genetic Mouse Models of CCA

Various genetic mouse models that portray the ample catalogue of mutations found in human CCA have been developed for the characterization of different stages of cholangiocarcinogenesis, ranging from the neoplastic transformation of normal liver or biliary cells to CCA progression and metastasis. In general, these models are based on three distinct genetic approaches: (1) somatic gene transfer into adult liver cells by hydrodynamic tail vein injection, liver electroporation, or adeno-associated virus (AAV) in vivo transduction (2) the manipulation of mouse embryonic stem cells to generate genetically-engineered mice, or (3) transplantation of pre-malignant cells, such as genetically engineered fetal liver cells or biliary organoids.

## 3. Somatic Gene Transfer Models

### 3.1. Hydrodynamic Tail Vein Injection (HTVI) Models

HTVI models are based on the delivery of plasmid DNA into hepatocytes by means of high-volume injection: controlled hydrodynamic pressure in capillaries enhances the permeability of endothelial and parenchymal cells, allowing DNA to enter the cells through the transient opening of “pores” in the plasma membrane (reviewed in [6]). Through the incorporation of the *Sleeping Beauty* transposon toolbox to the hydrodynamic injection technique, stable integration of transgenes can be achieved in several tissues [7,8]. Notably, the liver is particularly prone to plasmid DNA incorporation and HTVI efficiently targets up to 10% of liver cells. Therefore, several groups have adopted this technology for the generation of mouse models of liver carcinogenesis based on the introduction of genetic alterations found in the human counterparts [9].

HTVI models pose a series of advantages for in vivo studies. First, since only a fraction of hepatocytes is targeted by HTVI, normal and transformed cells coexist in the autochthonous environment, thus mimicking the human setting. Second, given that recipient mice are 6–8 weeks old, tumors develop in an adult organism, as is most commonly the case in humans. Third, many HTVI models form tumors very rapidly (1–2 months), thereby accelerating experimental readout. The main limitation of this method is the fact that HTVI delivers genes exclusively into hepatocytes of the pericentral region (zone 3 of the liver acinus). Therefore, a transdifferentiation stimulus, likely induced by the respective transgenic driver, is needed to facilitate CCA-like histologies, and studies aimed at investigating the role of hepatic stem cells or biliary epithelial cells to CCA carcinogenesis are not possible using this approach. Furthermore, all HTVI studies focus on CCA initiation, i.e., to understand the role of CCA-relevant genes in cancer formation, but not in progression and metastasis. Lastly, in contrast to what is frequently observed in the human situation, tumors in conventional HTVI models develop in the absence of an inflammatory microenvironment, which limits their applicability for studies of tumor–stroma crosstalk. Table 1 summarizes the HTVI models of CCA described to date, with some of the most representative and/or time-efficient models discussed in more detail.

#### 3.1.1. RAS-Driven HTVI Models

In 2005, the Largaespada’s lab provided the proof-of-concept that transposon technology can be used to study liver tumorigenesis [10]. In the first example of an HTVI liver cancer model, exogenous expression of *NRAS* oncogene (G12V) in *Arf*^−/−^ mice gave rise to an early onset of multiple nodular tumors 4 to 6 weeks after transfection. Tumor latency doubled in heterozygous *Arf*^+/−^ mice, highlighting the importance of the genetic background of recipient mice in liver carcinogenesis. Importantly, both HCC- and iCCA-like tumors arose in this model. Thus, the application of this liver cancer model to exclusively study CCA pathobiology is limited.

Another HTVI-based study modeling the consequences of RAS-driven carcinogenesis was later described by Zhang et al. Here, hydrodynamic injection of *NRAS (G12V)* and a constitutively active form of *AKT* (*myrAKT*) also yielded HCC and iCCA in mice 3 to 4 weeks after injection [14]. Interestingly, a latter version of the model incorporated the genetic ablation of FASN, the master regulator of fatty acid synthesis, whose expression is upregulated in human HCC samples but is frequently low in human iCCA specimens. Upon administration of a Cre recombinase transgene simultaneously with *NRAS* and *AKT* in *Fasn*^flox/flox^ mice, developing tumors exhibited nearly exclusively iCCA differentiation, highlighting the dependency of HCC—but not CCA—on de novo lipid synthesis [15].

#### 3.1.2. NOTCH-Driven HTVI Models

One of the most rapid models of iCCA, which displays human CCA features such as high mitotic activity, invasion of the surrounding liver parenchyma and necrosis, was reported in 2012. Based on the frequent activation of NOTCH signaling found in human CCA, Fan et al. overexpressed the intracellular domain of the NOTCH1 receptor (NICD) that shuttles from the membrane to the nucleus to function as a pro-oncogenic transcriptional regulator. Single administration of *NICD* transgene yielded CCA-like lesions exhibiting initial features of malignant transformation with a latency of 20 weeks. Notably, concomitant transfection of *NICD* and *AKT* significantly accelerated tumor development and led to formation of similar lesions with a latency of only 3.5 weeks that progressed rapidly and occupied most of the liver parenchyma by 5 weeks [12]. Paralleling this approach, transposon-based delivery of *NICD* and Cre-recombinase into genetically engineered mice harboring the latent mutant *Kras* allele (*Kras*^LSL-G12D^) gave rise to liver tumors closely resembling human CCA by 8 weeks post-transfection [17].

Upstream activation of NOTCH signaling can occur through the NOTCH ligand JAG1, which is frequently upregulated in human CCA. Further supporting the decisive effects of NOTCH signaling on cholangiocarcinogenesis, concomitant delivery of *myrAKT1* together with JAG1 induces exclusively iCCA formation, but not HCCs, by 8 weeks post-transfection [18].

A conceptually similar HTVI model took advantage of Yes-associated protein (YAP) overexpression. YAP, a major downstream effector of Hippo-signaling, is considered to act as an oncogenic driver in HCC and is also active in CCA [25]. Notably, YAP can positively regulate JAG1 expression [26], providing a direct link to NOTCH signaling. Concomitant *YAP* and *myrAKT* transgene overexpression led to activation of PI3K and RAS signaling, as well as an enhanced activity of the glycolytic pathway, resulting in early iCCA formation within 3 weeks post-transfection and death of tumor-bearing mice between 5.5 and 7.5 weeks. MTORC1 and mTORC2 are key downstream signaling components of AKT activation. Notably, while knockdown of mTORC1 significantly delayed tumor development, genetic abrogation of mTORC2 not only delayed tumor onset, but also led to the development of tumor lesions with hepatocellular -instead of cholangiocellular- differentiation [13].

As a second transposon-based HTVI model that featured YAP as a co-oncogenic driver, Li et al. simultaneously delivered active-mutant forms of *YAP* (S127A) and *PIK3CA* (H1047R) into the mouse liver, which resulted in the formation of liver tumors by approximately 12 to 13 weeks post-delivery. In this model, tumor histology ranged from HCC (40%) and iCCA (10%), to mixed HCC/iCCA (50%) [11]. Overall, the differences observed through overexpression of distinct elements within the same signaling pathways reinforce the idea that a precise selection of transgenes is critical to steer the development of appropriate HTVI models of CCA.

As a note of caution concerning the generalization of the reported latencies and histological presentations, it cannot be excluded that tumor kinetics as well as histological presentation may, in part, be dependent on the strain background and/or the gender of the “recipient” mouse.

#### 3.1.3. CRISPR-Cas9-Based HTVI Approaches

In recent years, the genome-editing field has been extraordinarily transformed by the bacterial RNA-guided clustered regularly interspaced palindromic repeat (CRISPR)-Cas9 system. The CRISPR-Cas9 technology provides an efficient method for somatic gene editing of cancer-related genes, both to generate functional knockouts (by the introduction of insertion or deletions-indels-in specific genes) and to recreate driver mutations (by homology-directed repair) [27]. Pioneering work by Xu et al. in 2014 illustrated that HTVI and CRISPR-Cas9 approaches could be successfully combined for the study of intrahepatic tumors. Hydrodynamic administration of single guide RNAs (sgRNAs) to target the TSGs *Tp53* and *Pten* in wild-type FVB mice originated tumors of biliary differentiation 3 months after injection, faithfully recapitulating the liver lesions found in adeno-Cre-activated, *Tp53*^flox/flox^; *Pten*^flox/flox^ mice [22].

One step further, the Rad lab took advantage of the CRISPR-Cas9 technology to perform a multiplexed in vivo mutagenesis screen in order to identify biologically relevant TSGs involved in liver cancer formation. A pool of 10 sgRNAs was injected together with a *Cre recombinase* transgene into *Alb-Cre; Kras*^LSL-G12D/+^ mice, resulting in the formation of multifocal liver tumors of HCC and iCCA histology 20 to 30 weeks after injection. Next generation sequencing (NGS) and comparative genomic hybridization (CGH) detected no off-target effects in CRISPR-Cas9 induced liver tumors, highlighting the accuracy of the gene editing strategy [21]. Given the wealth of genomic information from NGS efforts, this study provides the experimental platform to quickly and efficiently screen biological functions of newly identified CCA-related gene sets. Collectively, these two studies demonstrate that HTVI and CRISPR-Cas9 approaches are complementary tools for the development and investigation of CCA genetic models.

### 3.2. Liver Electroporation Models

Liver electroporation is an alternative technique for efficient hepatic gene delivery. Following laparotomy, the liver is exposed, plasmids are injected into the parenchyma, and an electric pulse is applied at the injection site. In contrast to HTVI approaches, which generally lead to the development of multiple tumor nodules across the liver, liver electroporation generates mostly focal tumors at the site of the electroporation.

A pioneer study by Gurlevik et al. demonstrated that *Sleeping Beauty*-mediated expression of *Kras*^G12V^ in *p53*^f/f^ mice efficiently leads to iCCA development through the transdifferentiation of hepatocytes. Additionally, this study highlighted the utility of the model to address adjuvant treatment strategies after R0 resection [24].

More recently, in 2018, Seehawer et al. reported significant differences in the histology of liver tumors formed when identical sets of oncogenes were introduced into hepatocytes by HTVI or liver electroporation techniques. Co-expression of oncogenic *Myc* and *Nras*^G12V^, or *Myc* and *AKT1* gave rise to multifocal liver carcinomas with HCC histology when introduced using HTVI, whereas, upon liver electroporation, the same combination of oncogenes resulted in focal iCCA or combined iCCA–HCC development. The authors went on to show that the surrounding hepatic microenvironment determines the cell fate of liver tumors, revealing that, in transformed hepatocytes containing the same oncogenic drivers, neighboring hepatocytes undergoing apoptosis induce the formation of HCC, while surrounding necroptotic hepatocytes determine CCA outgrowth [20]. Thus, this study highlights the importance of epigenetic cues from the tumor microenvironment in shaping the “phenotypic” differentiation of liver tumors.

### 3.3. Intrabiliary Transfection of DNA

As discussed earlier, the development of new CCA models based on the HTVI as well as the liver electroporation approach is compromised by the delivery of transgenes into hepatocytes, but not into cholangiocytes. To circumvent this limitation, a novel model that facilitates the direct transfection of biliary cells by intrabiliary injection coupled with lobar bile duct ligation was recently described by Yamada et al. Introduction of *AKT* and *YAP* transgenes into biliary cells led to CCA development in 20% of mice 10 weeks after in vivo transfection. Penetrance of tumor development could be further increased to 72% through an IL-6 dependent mechanism upon intraperitoneal injection of the biliary mitogen IL-33 [23]. The authors emphasized that tumor formation is not restricted to specific locations within the biliary tree. Therefore, despite photographs from the publication suggesting that most tumors arose as intrahepatic CCA, it appears likely that the approach could be adapted to generate extrahepatic CCAs (eCCAs). Although technically challenging, this model offers new opportunities for the generation of CCA mouse models by transposon-based gene delivery to the biliary epithelium and, considering the IL6-dependent increase in tumor formation, somewhat recapitulates the tumor-promoting role of inflammatory cues from the microenvironment.

## 4. Genetically Engineered Mouse (GEM) Models

GEM models are widely considered the most sophisticated animal models of human cancer. In these autochthonous cancer models, tumor initiation occurs de novo under the tight control of specific genetic modifications [28,29]. To date, multiple GEM models exist that faithfully recapitulate many histopathological features of human cancers, from early transformation stages to metastasis. Moreover, GEM models share molecular features with human cancers, as already illustrated in pioneering studies based on cross-species comparison analyses [30,31,32]. Since then, almost any GEMs, including those representative of CCA, have been investigated for their cellular and molecular similarities with their human cancer counterpart. For these reasons, GEM models have been incorporated as in vivo platforms for the rationale testing of standard-of-care chemotherapies and/or novel targeted agents [33].

GEM models of CCA developed to date incorporate many of the frequent oncogenic alterations found in humans [34,35,36,37,38,39], thus tightly mirroring the molecular events driving cholangiocarcinogenesis. A key advantage of these models is the development of tumors within the context of a complex organism. Additionally, the spectrum of preneoplastic, early and late tumor stages (including metastasis) in a highly reproducible setting enables the interrogation of critical aspects of carcinogenesis. However, the generation of new GEM strains requires complex and time-consuming breeding strategies, and tumor latency is relatively long in most cases, which translates into a significant economical investment. Unlike in humans, expression of the respective genetic alterations already occurs during embryogenesis in most non-inducible GEM models, although newly developed mouse strains aid to circumvent this problem. For instance, the incorporation of tamoxifen inducible, organ-specific Cre-recombinase-encoding alleles allows activation of latent alleles on demand in adult organisms. Further, hepatocyte-specific activation/incorporation of transgenes in GEM models can be easily achieved through incorporation of in vivo transfection techniques (HTVI and/or electroporation, described above), but also by using adeno-associated-virus-based AAV8 vectors. Intravenous injection of AAV8 vectors achieves efficient transgene delivery to hepatocytes, and additional specificity can be secured by using liver-specific promoters to drive transgene expression. Beyond the relative organ specificity, the use of endogenous promoters to drive transgene expression may more closely recapitulate physiological expression levels of the transgene in the host organism. Table 2 provides a summary of CCA GEM models reported to date, with a more detailed description of some key approaches found below.

### 4.1. Loss of Smad4 and Pten Tumor Suppressor Genes

In 2006, Xu et al. developed the first GEM model of CCA. Combined ablation of liver-specific *Pten* and *Smad4*, two mutations commonly found in human CCA [34,35,36], was obtained by crossing mice harboring the conditional alleles for each TSG to the Albumin Cre (*Alb-Cre*) strain [40]. The *Alb-Cre* strain allows recombination of the loxP sites in adult hepatocytes as well as in liver precursor cells of hepatocytes and cholangiocytes during embryonic development [58]. In the *Alb-Cre*; *Smad4*^flox/flox^; *Pten*^flox/flox^ model, hyperplastic foci arose from bile ducts at 2 months, and all mice developed iCCA between 4 and 7 months of age. Thus, the model preserves the gradual progression from bile duct hyperplasia and dysplasia to carcinoma in situ and invasive CCA that is frequently seen in the human situation. Similar to HTVI models, activation of the PI3K-AKT-mTOR pathway led to the expected phosphorylation of downstream targets, such as AKT, FOXO1, mTOR and GSK3β. Additionally, hyperactivation of p-ERK and overexpression of cyclin D1 was observed. Finally, the authors showed a reciprocal regulation of *Pten* and *Smad4* to maintain an expression balance to suppress CCA, thus highlighting the importance of the dual inhibition during CCA development in this model.

### 4.2. Simultaneous Activation of Kras and Deletion of Tp53

*KRAS* and *TP53* mutations are two of the most frequent genetic events in human CCA with prognostic implications for patients [37,39]. In 2012, pioneering studies from the Hezel lab reported the first GEM model with combined *Kras* activation and *Tp53* deletion [45]. In this model, *Kras**^LSL-G12D/+^*; *Tp53*^flox/flox^ mice were bred to the *Alb-Cre* mouse strain. Liver tumors developed with full penetrance in 9-week old mice, which succumbed to the disease with a mean survival of 19 weeks. Two thirds of the tumors histologically resembled stroma-rich iCCAs, while the remaining mice developed mixed HCC/iCCA or HCC at a similar frequency (17% each). The presence of premalignant biliary lesions (intraductal papillary neoplasms -IPNB- and Von Meyenburg complexes -VMC-), indicated that in this model tumors gradually evolve in a stepwise process that phenocopies the human situation, culminating in the progression to invasive carcinoma and the potential to metastasize to distal organs.

As Albumin-Cre is already active during embryonic development in bipotential hepatic progenitors, the cell of origin for iCCA formation in response to activation of *Kras* and *Tp53* remained unclear in this model. To further interrogate whether, in an adult organism, hepatocytes and/or cholangiocytes are amenable to Kras/p53 driven malignant transformation, the authors either directed expression of Cre-recombinase towards the biliary compartment by breeding the tamoxifen-inducible, *Sox9-Cre*^ERT2+^ allele into *Kras*^LSL-G12D/+^; *Tp53*^flox/flox^ mice, or injected *Kras*^LSL-G12D/+^; *Tp53*^flox/flox^ mice with an adeno-associated vector expressing Cre-recombinase under the control of the hepatocyte-specific thyroid-binding globulin promoter (AAV8-TBG-Cre) [59].

AAV-mediated expression of *KrasG12D* and loss of *Tp53* in the adult hepatocyte was only sufficient to drive tumorigenesis in this model when combined with an inflammatory stimulus by co-administration of DDC-diet, leading to development of iCCA, but also HCC with a similar incidence, as well as mixed iCCA/HCC tumors. In contrast, more rapid and mostly iCCA development was observed without an additional inflammatory stimulus when the transgenes were activated in the adult ductal compartment upon expression of *SOX9*-driven Cre-recombinase. Notably, evidence of tumors developing through pre-neoplastic stages was only found in the Sox-9 approach, but not upon DDC treatment [46].

In line with previous reports (e.g., [60]), these results confirm the functional role of *Kras* and *Tp53* in CCA and highlight the influence of liver inflammation on liver cancer formation by priming hepatocytes for oncogenic transformation.

### 4.3. Concurrent Activation of Kras and Abrogation of Pten

A third model incorporating oncogenic *Kras* was reported by Ikenoue et al. in 2016. Expression of mutant *Kras* and concomitant deletion of the TSG *Pten* was targeted to liver cells using the *Alb-Cre* mouse strain. Active *Kras^G12D^* synergized with homozygous *Pten* ablation to form multifocal, stroma-rich intrahepatic iCCA. On average, early hyperplastic biliary foci were detected around four weeks after birth, and mice died after 46 days, making the model one of the fastest CCA GEM models described to date using the *Alb-Cre* strain [42]. No metastatic lesions were described in this model. Notably, the authors reported that Kras activation in combination with heterozygous *Pten* deletion developed liver tumors with both hepatocyte and cholangiocyte differentiation, starting at 6 months of age, indicating a dosage-effect of Pten expression. Paralleling the approach by Hill et al. [46], the authors used two additional tamoxifen-regulatable GEM, *Alb-Cre*^ERT2+^ or *K19Cre*^ERT/+^ to activate Kras and delete Pten expression in the adult organism selectively in hepatocytes or cholangiocytes, respectively. However, in this particular model, no inflammatory additional stimulus was introduced. Interestingly, administration of tamoxifen to adult, 8-week old *Alb-Cre*^ERT2+^; *Kras*^LSL-G12D^; *Pten*^flox/flox^ mice lead to the development of HCC and HCC-precursor lesions, but not iCCA, whereas early postnatal tamoxifen injection on day 10 resulted in iCCA. Notably, lineage tracing revealed that, on postnatal day 10, *Alb*-*Cre* was still active in biliary cells and, together with the observation that *K19Cre*^ERT/+^; *Kras*^LSL-G12D^; *Pten*^flox/flox^ mice gave rise to pre-malignant papillary ductal lesions in periportal areas, these findings suggest that CCA arises from cholangiocytes in these models. In support of these results, a similar GEM model was recently published by Lin et al., in which most of the previous findings are mirrored [43].

### 4.4. Mutations in the IDH Genes 

Mutations in *IDH1* and *2* genes occur in approximately 20% of human CCAs [61,62]. While wild-type IDH is part of the TCA cycle and converts isocitrate to α-Ketoglutarate, mutant IDH acquires a neomorphic enzyme activity that results in the aberrant production of 2 Hydroxyglutarate (2-HG) from α-Ketoglutarate. Intracellular accumulation of 2-HG blocks hepatocyte lineage progression and inhibits multiple enzymes that utilize α-Ketoglutarate as a co-enzyme and are involved in epigenetic regulation [63]. In 2014, Saha et al. developed latent mutant *Idh2* (*Idh2*^LSL-R172K^) mice that were bred to the *Kras*^LSL-G12D^ and *Alb-Cre* strains to determine the impact of IDH mutations on cholangiocarcinogenesis in the context of mutant Kras. Multifocal liver masses resembling iCCA that exhibited invasive growth patterns and metastatic potential formed in mice ranging from 33 to 58 weeks of age. Oval cell expansion and the presence of biliary intraepithelial neoplasia-like lesions adjacent to tumor foci suggested that the tumors might evolve from such preceding preneoplastic stages [48]. Recently, positive phase III data for CCA patients harboring *IDH* mutations were reported for the IDH inhibitor ivosidenib, emphasizing the direct relevance of such preclinical platforms for “co-clinical trials” to establish mechanisms and biomarkers of response and resistance to the molecularly targeted inhibitors.

### 4.5. Activation of the Notch Pathway 

Conceptually similar to the HTVI models mentioned above, Zender et al. investigated the influence of the NOTCH pathway in CCA development using a GEM model. When NICD was expressed under the regulation of *Alb-Cre,* initial features of malignant transformations were present within the liver by the age of 8 months and gave rise to iCCA-like tumors upon transplantation of primary tissues into recipient mice [49]. In contrast, in the HTVI-based NICD overexpression model, tumors formed 20 weeks after introduction of the transgene [12]. The reduced tumor latency might in part be attributable to different levels of transgene expression, an alternate “cell of origin” targeted with the oncogenic driver, as well as strain differences.

### 4.6. Mitochondrial Dysfunction

Mitochondrial dysfunction leads to increased ROS levels and severe liver injury. In a model developed by Yuan et al., this inflammatory environment was induced by the disruption of mitochondrial protein homeostasis in *Hspd1*^flox/flox^ mice crossed to the *Alb-Cre* strain. Within the severely damaged liver, focal areas of regenerating hepatocytes and cholangiocytes emerged, the latter showing resemblance to human biliary intraepithelial neoplasia. The transgenic animals succumbed to severe liver injury prior to development of fully established iCCA. However, transplantation of liver tissues gave rise to transplantable tumors that histologically showed CCA features.

Notably, most of the regenerating foci observed in the primary livers retained Hspd1 expression, indicating that the lesions evolved in the presence of the injured microenvironment but not through cell-autonomous effects resulting from *Hspd1* deletion. Indeed, the authors show that the biliary reaction is the consequence of a Jnk-mediated response to Tumor necrosis factor released from Kupffer cells, triggered by hepatic mitochondrial dysfunction and oxidative stress. While this model does not allow for long-term experiments, due to the premature death of the transgenic animals, it uniquely highlights the fundamental role of the microenvironment in the setting of chronic liver injury, to create a “pre-carcinogenic niche” that promotes tumor development. The non-cell autonomous mechanism that characterizes the development of CCA-precursor lesions in this model contrasts with the cell-autonomous effects primarily involved in oncogene-driven GEM models.

### 4.7. Carcinogen-Treated GEM Models 

Administration of carcinogens partially mimics the consequences of an inflammatory environment, a frequent underlying risk factor in the development of human CCA. Integrating carcinogen-induced liver injury and genetically “sensitized” murine models, Farazi et al. combined genetic deletion of *Tp53* with administration of the biliary toxin CCL_4_. In this model, iCCAs developed in the setting of a fibro-inflammatory microenvironment with 50% penetrance in *Tp53*^−/−^ mice and with 18% in heterozygous *Tp53*^+/−^ mice [57].

A decade later, Guest et al. found that administration of the hepatocarcinogen thioacetamide (TAA) to mice lacking *T**p53* in the cholangiocyte compartment, led to the development of iCCA with long latency (>26 weeks) and incomplete penetrance [56]. It is highly likely that co-treatment with carcinogens would also accelerate tumorigenesis in related genetic models of CCA, potentially allowing for a dose-dependent titration of tumor development and a more faithful recapitulation of the pro-inflammatory microenvironment that is considered to influence tumor pathobiology and treatment response.

### 4.8. Extrahepatic CCA

So far, the different GEM models described above are models of iCCA. In 2017, Nakagawa et al. described the first GEM model of eCCA. Although the *K19*^CreERT^ mouse strain achieves similar recombination efficiencies across the different portions of the biliary tree, activation of the *Kras* oncogene and concomitant deletion of the *Tgfβr2* and *Cdh1 (E-cadherin)* in adult K19 positive cells led to CCA development predominantly in the extrahepatic and perihilar bile ducts. The authors provide evidence that the peribiliary glands, located within the walls of the extrahepatic bile ducts, are the site of malignant transformation in this model, resulting in early-onset aggressive tumors metastatic to the regional lymph nodes. Interestingly, the authors show that IL-33, likely released from dying cholangiocytes due to E-cadherin loss, exerts a pro-proliferative stimulus on biliary epithelial cells, thereby actively contributing to malignant transformation. Since *K19*^CreERT^ is active in the liver and in the lung, most animals die within 4 weeks after tamoxifen administration, due to liver and/or respiratory failure. Therefore, the model is not suitable for long-term experiments. However, the authors show that some of the crucial features of the model can be recapitulated and further followed up using organoids derived from the multiallelic mice [52].

This study highlights how the proper combination of genetic alterations can guide CCA development towards distinct disease subtypes. In conjunction with the *Hspd*^flox/flox^ model of iCCA, these findings further illustrate how genetic events can profoundly influence the surrounding microenvironment to create a pro-inflammatory context that, in turn, exerts deleterious effects on the epithelial liver cells.

## 5. Allograft Models

As an alternative to conventional GEM models, implantation of pre-malignant liver cell populations into recipient mice emerged as a time-efficient approach that circumvents complex and time-consuming breeding schemes.

### 5.1. Liver Implantation of Liver Progenitor Cells (LPCs)

In 2013, we described an allograft CCA mouse model based on the orthotopic implantation of fetal LPCs isolated from *Alb-Cre*; *Kras*^LSL-G12D^; *Tp53*^LSL-R172H/flox^ mice, resulting in the death of recipient animals approximately 3 months after inoculation. Fetal liver cells can be easily further genetically modified, for example through the introduction of retroviral vectors that overexpress oncogene-encoding cDNA cassettes or encode for shRNAs to mimic TSG loss by RNA interference [53].

### 5.2. Liver Implantations of CCA Organoids

Recently, liver organoid technology has evolved as a novel system to model hepatobiliary malignancies [54]. In contrast to LPCs, murine adult liver-derived three-dimensional (3D) organoids can be propagated long-term in vitro and used on-demand, thereby abrogating the need for repetitive isolations of primary cells, while retaining the genetic flexibility of the LPC approach. Organoids can be rapidly engineered using various gene-editing strategies, and transplantation into the livers of syngeneic mice enables tumor growth in an autochthonous and immune-competent microenvironment. Notably, both LPCs and liver organoids retain their plasticity to give rise to both CCA and HCC-like tumors, depending on the genetic context.

## 6. Cell Plasticity Leading to CCA: Evidence from Mouse Models

The lineage tracing approach to identify the cell of origin of a given cancer has been extensively utilized in the last decade [64,65]. This approach has been also successfully applied to CCA, with the purpose of unravelling whether these tumors arise from hepatic progenitor cells (HpSC), cholangiocytes, or hepatocytes. Some of the landmark studies on this topic are summarized here. Specifically, a HpSC origin of CCA was suggested in the AlbCre/N2ICD (Notch 2 intracellular domain) model treated with the hepatocarcinogen diethylnitrosamine (DEN) [66] as well as in mice overexpressing the active form of Notch1 (AFP-NICD mice) [67]. However, the HpSC origin of CCA has been questioned by tracing the thyroxine-binding globulin in mice subjected to diethylnitrosamine (DEN) and multiple injections of CCl4 or TCPOBOP [68]. Similarly, using a cholangiocyte-lineage tracing system (CK19-lineage tracing) to target p53 loss in biliary epithelia, it was found that the biliary epithelium is in fact the specific target of transformation and origin of iCCA in this model [69]. Nonetheless, the mature hepatocyte has also been identified as the cell of origin of CCA in lineage-tracing studies tracing albumin [51] or transthyretin positive parenchyma cells [12]. In the latter study, the origin of CCA by mature hepatocytes was further confirmed by electron microscopy findings.

The molecular mechanisms driving the transdifferentiation process in the liver, i.e., the capability of a given cell type to convert into a different one, remain to be better elucidated in cholangiocarcinogenesis. These mechanisms presumably implicate the involvement of multiple genetic and epigenetic events, ultimately leading to the activation and interplay of several signaling pathways. Emerging evidence implies a crucial role of the Hippo and Notch pathways in the hepatocyte-to-malignant cholangiocyte conversion [11,12,14,18,51]. In particular, it has been shown that iCCA developing in AKT and Yap co-expressing mice derive from mature hepatocyte, and this process is strictly dependent on the canonical Notch2 signaling cascade [13]. Indeed, deletion of *Notch2* in AKT/Yap-induced tumors switched the phenotype from iCCA to hepatocellular lesions (adenomas or HCC). In contrast, inactivation of *Notch1* in hepatocytes did not lead to any significant histomorphological modification of the original lesions (iCCA) in this model, identifying Notch2 as the major determinant of hepatocyte-derived iCCA [69]. Obviously, further studies are necessary for a better understanding of the transdifferentiation process in CCA.

Although the summarized data are highly helpful to identify the cell of origin of CCA and, thus, to better elucidate the role(s) of the different cell types in cholangiocarcinogenesis, their relevance for the human disease remains to be addressed. Presumably, as shown by the various experimental models, the cell of origin of human CCA may also vary, depending on the oncogenes involved, the type, degree and duration of the pro-oncogenic insults, the status of the liver, etc. In support of this hypothesis, it is worthwhile to mention a paper by Holczbauer et al. [60]. In this study, the authors found that, irrespective of origin, HpSc, lineage-committed hepatoblasts, and differentiated adult hepatocytes transduced with transgenes encoding oncogenic H-Ras and SV40LT acquired markers of stemness, side populations, and self-renewal capacity in vitro. In addition, these transduced cells were also able to form a wide spectrum of liver tumors in vivo, such as CCA and hepatocellular carcinoma (HCC), regardless of the cell of origin. Furthermore, the molecular profiles of each tumor subtype (CCA, HCC, and undifferentiated/mesenchymal-like tumors) were highly similar irrespective of the cell of origin, implying that potentially any cell type in the mouse hepatic lineage might undergo oncogenic reprogramming into a “virtual” HpSC and drive development of CCA and/or other tumor types [60].

## 7. Summary

Genetic mouse models recapitulate many hallmarks of cholangiocarcinogenesis that cannot be modeled using in vitro systems or, even, by xenotransplantation of human CCA cell lines or patient-derived tissue (PDXs) into immunodeficient mice, two alternative in vivo models that have been the focus of comprehensive reviews elsewhere [70,71]. Table 3 highlights the main features of the different in vivo approaches and is intended for the selection of the most adequate model for each study. A particular drawback of any mouse model is the intrinsically high cost of in vivo experiments due to mice number per study, husbandry and maintenance. This issue becomes especially critical in the case of GEM, where breeding schemes can be lengthy, or in those genetic models where tumor latency is significantly longer than in graft models (>two months).

A key advantage of genetic mouse models over human cell/tissue-xenografted mouse models, however, is the suitability of murine models to investigate tumor development and tumor-stroma crosstalk within the context of an immunocompetent microenvironment. In traditional transgenic approaches, tumor development can be followed through gradual changes within the epithelial tumor cell compartment, along with a remodeling of the surrounding stroma, thereby allowing researchers to also investigate the early processes of iCCA development within an otherwise intact microenvironment. This “stepwise” tumorigenesis may be only incompletely mirrored in somatic gene transfer and murine allograft model systems. However, these models bear the genetic flexibility to rapidly incorporate additional genetic alterations to functionally annotate potential cancer driver genes in vivo and to characterize them not only regarding their transformative potential, but also concerning their effects on the tumor microenvironment. Furthermore, several models also depend on the consequences of an inflammatory stimulus on tumorigenesis that is derived from the microenvironment itself. 

Clearly, no single model will be capable to adequately address all the questions related to cholangiocarcinogenesis. Although the models developed thus far have considerably helped us to better understand important aspects of cholangiocarcinogenesis, at the same time, they have also clearly demonstrated how complex liver tumor formation is. For instance, several of the models can give rise to the full spectrum of liver cancer, from CCA to HCC to mixed tumors. In some of the models, those differences can be traced back to certain engineered genetic alterations within the epithelial tumor cell compartment, to gene dosage, and to microenvironmental cues that influence the determination of liver tumorigenesis into biliary or hepatocyte cell lineage differentiation. Similarly, allograft models using primary liver progenitors or organoids can also be directed towards the different lineages of primary liver cancers, depending on the respective cancer drives. When these models are taken seriously, one has to acknowledge that a “dogmatic” approach that tries to establish that “one cell of origin” for CCA may be a misguiding simplification of highly complex mechanisms that guide cellular transdifferentiation and/or neoplastic transformation in the course of liver tumor development.

While the observations that some models lead, in part, to tumors of hepatocellular or mixed differentiation suggests an avenue to their applicability towards understanding tumor “lineage” differentiation, it also indicates how important a thorough a priori characterization of these model systems is in order to adequately use them in translational research. Considering the devastating prognosis and the lack of durable responses, we need accurate pre-clinical systems that can recapitulate the challenges encountered when CCA patients are treated in clinical practice; although effective in selected patient subgroups, precision oncology is hampered by early relapse or insufficient response, despite the presence of targetable genetic lesions. Genetically flexible model systems have the potential to help us to improve our understanding of primary and secondary resistance mechanisms that influence the response to targeted therapies. In addition, these immune-competent in vivo systems may also help us to understand why immune-oncology approaches have thus far failed in CCA and guide us in the implementation of novel therapeutic concepts.

## 8. Concluding Remarks

Although the development of an ideal mouse model for the study of CCA is unrealistic, in part due to the genetic and histological complexity of this tumor, to date, the spectrum of genetic mouse models offers the research community an unprecedented opportunity to elucidate CCA pathophysiology. Currently, the therapeutic portfolio for CCA patients is finally starting to expand considerably: precision medicine is on the verge of becoming an integral component of CCA treatment, and immune-oncology approaches are actively pursued. This development emphasizes the need for immune-proficient, translational model systems in which cancer cell intrinsic and extrinsic mechanisms can be adequately addressed. Based on insights from current clinical trials and integrative molecular profiling studies that imply the existence of several molecular CCA subgroups, it is likely that, within the coming years, the repertory of genetic mouse models will further expand. Building on the “groundwork” that has been laid by the existing approaches, more diversified in vivo systems will be developed, and serve as pre-clinical platforms to address the prevailing questions, from the characterization of the initial steps during cholangiocarcinogenesis, the dissection of histopathological and molecular features, to response and resistance to novel therapeutics.

## Figures and Tables

**Table 1 cancers-11-01868-t001:** Genetic models based on liver transgene delivery. Summary of CCA models based on transgene delivery via HTVI or liver electroporation. iCCA: intrahepatic CCA; eCCA: extrahepatic CCA; HCC: hepatocellular carcinoma.

Somatic Gene Transfer Delivery Models		
Transgene and Mouse Genetic Background	Key Features	Reference
*NRasV12* in *Ink4A/Arf*^−/−^ mice	Development of mixed HCC/iCCA 4–6 weeks after HTVI	[10]
*Yap* and *PI3KCA* in wt mice	HCC (~40%), CCA (~10%), and mixed HCC/iCCA (~50%) formation. iCCAs cover ~80% of the liver parenchyma within 12–13 weeks post-HTVI	[11]
*NICD1* in wt mice	HTVI induces formation of cystic cholangiocellular tumors resembling human biliary cystadenomas by 20 weeks	[12]
*NICD1* and *AKT* in wt mice	Quick formation of iCCA (4.5 weeks after HTVI) with signs of malignancy such as high content of mitotic figures, necrosis and invasion of the liver parenchyma	[12]
*MyrAKT* and *YAP* (S127A) in wt mice	Quick (3 weeks post-HTVI) model of iCCA	[13]
*MyrAKT* and *NRAS* (V12D) in wt mice	Quick (3 weeks post-HTVI) model of mixed HCC/iCCA	[14,15,16]
*NICD* in *Kras*^LSL-G12D^ mice	Tumors of ductal and cystic morphology with variable amount of desmoplastic stroma invading the surrounding hepatocellular parenchyma 8 weeks after HTVI	[17]
Overexpression of *Jag1* and *AKT* in wt mice	Tumors of solid, ductular or cystic differentiation with some stroma development 8 weeks after HTVI. Exclusively iCCA	[18]
*NICD*, *YAP* and *Tp53* shRNA	HCCs or iCCAs form 6 weeks post-HTVI in response to Wnt and Notch signaling respectively Nestin-positive progenitor-like cells (from de-differentiated mature hepatocytes) upon *Tp53* loss	[19]
*Myc* and *Nras* or *Myc* and *AKT*	HTVI delivery induces HCC, while delivery of the same oncogenes by electroporation forms stroma-rich iCCA (both by models develop by 4 weeks after transgene delivery)	[20]
Cas9, Cre and sgRNAs in *Kras*^LSL-G12D/+^mice	Formation of HCC and iCCA 20–30 weeks after delivery via HTVI	[21]
Cas9 and sgRNAs to *Pten* and *Tp53*	Liver tumors featuring bile duct differentiation markers (CK19 positive) by 12 weeks post-HTVI	[22]
*AKT* and *YAP* plus IL-33 intraperitoneal injection	Overexpression in biliary cells induces CCA formation along the biliary system 10 weeks after HTVI. IL-33 increases penetrance. Likely technically adaptable for eCCA formation.	[23]
*KRAS^G12V^* in *Tp53*^f/f^	Quick (3–5 weeks) model based on liver electroporation of *KRAS* (G12V) induces single nodule iCCA formation	[24]

**Table 2 cancers-11-01868-t002:** Genetically engineered mice (GEM) and GEM-derived models. Summary of GEM models and additional mouse models developed based on existing GEM. iCCA: intrahepatic CCA; eCCA: extrahepatic CCA; HCC: hepatocellular carcinoma.

GEM and GEM-Derived Models		
Genetic Strategy	Key Features	Reference
*Alb-Cre*, *Smad4*^flox/flox^, *Pten*^flox/flox^	Liver progenitor cells (LPCs) iCCA development involving a multistep sequence including hyperplasia, dysplasia, carcinoma in situ and well-established CCA. Tumor latency: 4–7 months	[40]
*AhCre*^ERT^, *Kras*^V12/+^, *Pten*^flox/flox^	Oncogenic events in biliary epithelium yield multifocal non-invasive papillary neoplasms in the intrahepatic biliary tract involving both major interlobular bile ducts and small bile duct radicles in portal tracts. Mice survive up to 43 days	[41]
*Alb-Cre*, *Kras*^LSL-G12D/+^, *Pten*^flox/flox^	Highly invasive and desmoplastic CCA originated from LPCs, with a glandular morphology that resembles well-differentiated human CCA. Tumors form by 4 weeks	[42,43]
*Alfp-Cre*; *Tp53*^flox/flox^	Loss of p53 in LPCs leads to formation of advanced HCC and iCCA. Tumor formation in 14–20-month-old mice	[44]
*Alb-Cre*; *Kras*^LSL-G12D/+^; *Tp53*^flox/flox^	Tumors of iCCA (66%), HCC (17%) and HCC/iCCA (17%) histology. iCCA with multistage progression including stroma-rich tumors, premalignant biliary lesions (IPNB and VMC) and metastasis. Tumors form as early as 9 weeks	[45]
*Sox9-Cre*^ERT2+^; *Kras*^LSL-G12D/+^, *Tp53*^flox/flox^	Development of iCCA tumors accompanied by adjacent extensive ductular reactions and desmoplasia, as well as areas resembling biliary intraepithelial neoplasia (BIN). Tumor latency: 30 weeks	[46]
*Kras*^LSL-G12D/+^; *Tp53*^flox/flox^	Mice infected with an adeno-associated virus expressing Cre (AAV-TBG-Cre) develop iCCA (40%), HCC (40%), mixed HCC/iCCA (20%). Tumors form between 12- and 66-weeks post-injection	[46]
*Alb-Cre*; *Kras*^LSL-G12D/+^; *Fbxw7*^LSL-R468C/LSL-R468C^	Dysplastic dust-like structures surrounded by fibrosis (at 8 months, some heterozygous Fbxw7^LSL-R468C^ mice show bile duct dilation and hyperplasia).	[47]
*Alb-Cre*; *IDH2*^LSL-R172K^; *Kras*^LSL-G12D^	All mice develop multifocal liver masses of iCCA histology. Tumor latency: 33–58 weeks	[48]
*Alb-Cre*; *NotchICD*	Transplantation of liver cells from 8 months-old mice in immunodeficient animals produces iCCA, likely derived from progenitor cells. Tumor latency: 8 months	[49]
*Alb-Cre*; *Tp53*^flox/flox^; *NotchICD*	iCCA with abortive glandular pattern (moderate to high pleomorphic nuclei with some atypical mitoses) developed along with a dense fibrous tissue with inflammatory cells. Tumors form by 6 months	[50]
*Alb-Cre*; *Hspd1*^flox/flox^	Lesions of cholangiocellular histology, characterized by irregular glands, loss of polarity, multilayering of cells, and frequent mitosis resembling human BIN, form by 4–6 weeks	[51]
*K19*^CreER^; *Kras*^LSL-G12D/+^; *Tgfβr2*^flox/flox^ and *Cdh1*^flox/flox^	Swollen gallbladder including invasive periductal infiltrating type eCCA metastasizing to lymphatic vessels and a prominently thickened EHBD wall. Mice die at 4 weeks of age	[52]
**GEM-based allograft models**		
Bipotent liver progenitor cells (LPCs) from *Alb-Cre*; *Kras*^LSL-G12D^; *Tp53*^LSL-R172H/flox^ mice +/− FIG-ROS fusion	Implanted LPCs give rise to liver tumors with advanced iCCA features. Median overall survival of *Alb-Cre*; *Kras*^LSL-G12D^; *Tp53*^LSL-R172H/flox^ tumor-bearing mice: 79 days	[53]
Bipotent or cholangiocytic progenitor cells or hepatocytes from *Tp53*^−/−^ mice	Tumors with mixed hepatocytic and cholangiocytic differentiation embedded in a prominent stroma. Tumors form between 30 and 120 days after cell inoculation	[44]
Adult liver organoids from *Kras*^LSL-G12D^*; Tp53*^flox/flox^ mice after Cre-activation in vitro	Kras driven tumors lead to CCA formation, while expression of c-Myc in wild-type organoids drives HCC formation. Tumor latency for Cre- activated *Kras*^LSL-G12D^ *Tp53*^flox/flox^: 6–8 weeks	[54]
**Carcinogen-treated GEM models**		
*Alb-Cre*^ERT2^; *R26*^RlacZ/+^ and *Ck19-Cre*^ERT2^; *R26*^RlacZ/+^ mice treated with TAA	Liver cirrhosis containing cells with the typical histology of iCCA	[55]
*Ck19-Cre*^ERT/eYFP^; *Tp53*^flox/flox^ mice treated with TAA	Formation of multifocal invasive iCCA. Tumors appear after 26 weeks	[56]
*Tp53*^−/−^ mice treated with CCl4	Development of injury/necrosis, proliferation and fibrosis in bile duct after 4 months of treatment	[57]

**Table 3 cancers-11-01868-t003:** Advantages and disadvantages of in vivo mouse models. Features were rate from best (+++) to worst (−). NA: not available for assessment.

Mouse Model	Somatic Gene Transfer	GEM	Implantation of Mouse Cells	CCA Cells-Based Xenografts	Patient-Derived Xenografts
Time for tumor development	++	±	+++	+++	+
Resource consumption	+	±	++	++	+
Technical training	±	++	++	++	±
Ease of maintenance	+++	+	+++	+++	±
Success rate of initiation	+++	+++	+++	++	+
Retention of phenotypic features	++	+++	++	±	+
Retention of genetic features	+++	+++	+++	++	+++
Representation of genetic spectrum	++	+++	++	++	+
Amenable to genetic modification	++	++	+++	+++	+
Physiological expression of transgene	±	+++	++	++	++
Matched normal controls	+++	+++	+	−	±
Representation of early stages of carcinogenesis	+	+++	+	−	−
Cell of origin	+	++	+++	−	−
Tumor progression	+++	+++	+++	++	++
Tumor-stroma interactions	++	+++	++	±	+
Immune microenvironment	++	+++	++	−	±
Chemo-/targeted therapies studies	+++	+++	+++	++	+++
Immuno-Oncology studies	++	+++	+++	−	+

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
