# Peer review of "Genetic Mouse Models as In Vivo Tools for Cholangiocarcinoma Research"

_cancers, 2019, doi:10.3390/cancers11121868_

Round 1
Reviewer 1 Report
Authors answered all my questions and now the paper has become more sophisticated. I don't have any additional comments or concerns for this manuscript.
Reviewer 2 Report
In my opinion the manuscuscript is accepatable in this revised form
This manuscript is a resubmission of an earlier submission. The following is a list of the peer review reports and author responses from that submission.
Round 1
Reviewer 1 Report
Although cholangiocarcinoma is a fatal disease, its mechanism is not well understood, and establishment of animal models that develop cholangiocarcinoma is important for clarification of its mechasnim. This review presents three murine models that have previously been reported to develop cholangiocarcinoma. The classification of cholangiocarcinoma mouse models used in the review is appropriate, and most of the previous reports seem to be covered. In order to make the paper even better, the following points should be corrected. Major point 1.There is no description about AAV-Cre based model. The similarity and difference with HTVi or Electroporation model and AAV-Cre baed model should be described. 2.There is a model that can be HCC-CCC mixed and a model that can only ccc, but there is no mention of the difference and usefulness for future research. 3.The origin of CCC is divided into hepatocyte, cholangiocyte, and progeniter cell. Is there any difference in CCC that can be produced by each origin? 4.In this paper, as a limitation of HTVi model mice, it is described that gene transduction is carried out only in hepatocytes in HTVi, leading to the development of cholangiocarcinoma via Transdifferentiation. How has it been demonstrated that bile duct cells or liver progenitor cells have not resulted in the development of cholangiocarcinoma? In addition, please summarize the known mechanisms of transdifferentiation from hepatocytes to bile duct cells during CCC formation.
Author Response
Response to reviewers’ comments for Erice et. al “GENETIC MOUSE MODELS AS IN VIVO TOOLS FOR CHOLANGIOCARCINOMA RESEARCH’’ ([Cancers] Manuscript ID: cancers-613344)
We thank the three reviewers for their detailed and helpful comments. We have now addressed the issues raised with significant changes to the manuscript where appropriate. We are confident that the revised version of the manuscript has been significantly improved and that all the comments have been properly addressed.
REVIEWER 1:
Although cholangiocarcinoma is a fatal disease, its mechanism is not well understood, and establishment of animal models that develop cholangiocarcinoma is important for clarification of its mechanism. This review presents three murine models that have previously been reported to develop cholangiocarcinoma. The classification of cholangiocarcinoma mouse models used in the review is appropriate, and most of the previous reports seem to be covered. In order to make the paper even better, the following points should be corrected.
Major point
There is no description about AAV-Cre based model. The similarity and difference with HTVi or Electroporation model and AAV-Cre based model should be described.
We thank the reviewer for bringing up this point. We have now included a more detailed explanation of the AAV-approach in the paragraph where the KrasLSL-G12D/+; Tp53flox/flox model transduced with the AAV-Cre was originally described.
There is a model that can be HCC-CCC mixed and a model that can only ccc, but there is no mention of the difference and usefulness for future research.
The reviewer raises an important point. Indeed, several models are capable of giving rise to the full spectrum of liver tumors. While this is, on the one hand, a problem when a strict CCA model is wanted, for example for pre-clinical drug studies, it also opens new avenues for a better understanding of the complex mechanisms that lead to cellular fate determination in liver cancers. Indeed, several authors (e.g. Ikenoue et al, Sci Rep 2016; Li et al, Hepatology 2016; Saborowski et al, Hepatol. Communications 2019) have described how the modification of the genetic context within the epithelial tumor cells can steer the development of tumor phenotypes towards different lineages, using both traditional as well as allograft approaches. Most recently, it has been demonstrated that also the microenvironment created by the experimental approach (HTVI vs liver electroporation) can fundamentally contribute to the differentiation of the resulting tumors towards HCC- or CCA, despite the use of the same oncogenic drivers (Seehawer et al, Nature 2018).
We have now emphasized on these observations, wherever applicable, in the description of the respective models within the main text. In addition, we are now discussing the “promise and perils” of different liver tumor differentiations in the summary of this article.
The origin of CCC is divided into hepatocyte, cholangiocyte, and progenitor cell. Is there any difference in CCC that can be produced by each origin?
We appreciate the reviewer’s insightful question. To our knowledge, there has been no formal side-by-side comparison to respond to this open question in the CCA field at least in humans. Nonetheless, pathological evaluation and molecular profiling of human tumors indicate that CCA can be divided into several subgroups, which is also underscored by the distinctly different mutational landscape of intra- vs extrahepatic CCA, as well as gallbladder cancers (e.g. Jusakul et al, Cancer Discovery 2017; Xue et al, Cancer Cell 2019). This heterogeneity suggests that not only a single cell has the potential to give rise to CCA in humans. As mentioned in our previous answer, murine models systems can, in part, recapitulate this “plasticity”, and, as reviewed in this manuscript, thus far histologically accurate CCA-models expressing classical biliary lineage markers have been successfully derived from hepatocytes, biliary epithelial cells, as well as progenitor cells.
For example, Hill et al (Cancer Res, 2018) report that while cholangiocytes are prone to transformation by Kras activation and Tp53 loss, hepatocytes are refractory to the combined oncogenic insult, but can undergo transdifferentiation and transformation when the genetic event is combined with an inflammatory stimulus. In addition, when using fetal liver progenitor cells but also organoids, which are frequently regarded to be derived from adult progenitor-like cell types, specific combinations of oncogenic events can lead to different liver cancer histologies (HCC vs CCA), as exemplified by our recent work (Saborowski et al, PNAS 2014 and Hepatology Communications, 2019).
On the other hand, once the preneoplastic and/or neoplastic lesion is formed, the molecular features of such a lesions might be independent from the cell of origin. Although this important issue requires further validation, emerging evidence supports the latter hypothesis. Indeed, it has been previously shown that similar liver tumors developed in terms of morphologic and molecular features (with the only major difference represented by the tumor latency period) from mouse primary hepatic progenitor cells, lineage-committed hepatoblasts, and differentiated adult hepatocytes transfected with oncogenic H-Ras and SV40LT (Holczbauer et al, Gastroenterology, 2013). In accordance with these findings, it has been found that the histopathological features and the microarray transcriptomic profiles of the AKT/Yap/IL-33 model, obtained by the intra-biliary transduction of AKT and Yap coupled with bile duct ligation followed by IL-33 administration, closely resembled those detected in AKT/Yap mice, obtained by the transfection of AKT and Yap in mature hepatocytes by hydrodynamic tail vein injection (Wang et al, Journal of Hepatology, 2019). This issue has been discussed in a new section of the manuscript (‘‘Cell plasticity leading to CCA: evidence from mouse models’’).
In this paper, as a limitation of HTVi model mice, it is described that gene transduction is carried out only in hepatocytes in HTVi, leading to the development of cholangiocarcinoma via transdifferentiation. How has it been demonstrated that bile duct cells or liver progenitor cells have not resulted in the development of cholangiocarcinoma?
In addition, please summarize the known mechanisms of transdifferentiation from hepatocytes to bile duct cells during CCC formation.
We agree with the Reviewer that a major limitation of the HTVi is in fact the targeting of only mature hepatocytes. This has been convincingly proven using lineage tracing approaches (described in the text). The final conclusion that mature hepatocytes are the targets of HTVi is also based on anatomical reasons. Indeed, if we assume that iCCA derive from progenitor cells using HTVi, it would mean that the injected plasmids bypass the liver acinus against the sinusoidal blood flow to reach the utmost periportal area, then transfect progenitor cells without being incorporated by hepatocytes along the way. This hypothesis is highly unlikely since it contradicts the physiologic principle of hydrodynamic gene delivery, known to target nearly exclusively hepatocytes located in acinar zone 3 (i.e., close to the hepatic venule). Hypothetically, small amounts of plasmids might reach the canals of Hering and be incorporated by progenitor cells. However, cells transfected by HTVi are fully differentiated hepatocytes located in zone 3 and, thus, preneoplastic lesions developed always in zone 3 vein proximity. In accordance with this assumption, it has been demonstrated that the affected single cells in AKT/Notch1 mice are never located in zone 1 but always in zone 3 of the liver acinus (Fan et al, Journal of Clinical Investigation, 2012). In the same study, it was shown by way of electron microscopy the presence of tight junctions between transfected cells and normal hepatocytes, thus implying their hepatocellular nature.
The molecular mechanisms driving the transdifferentiation process in the liver, i.e. the capability of a given cell type to convert into a different one, remains to be better elucidated in cholangiocarcinogenesis. These mechanisms presumably implicate the involvement of multiple genetic and epigenetic events, ultimately leading to the activation and interplay of several signaling pathways. Emerging evidence implies a crucial role of the Hippo and Notch pathways in the hepatocyte-to-malignant cholangiocyte conversion (11, 13, 15, 19, 62). In particular, it has been shown that iCCA developing in AKT and Yap co-expressing mice (18) derive from mature hepatocyte and this process is strictly dependent on the canonical Notch2 signaling cascade. Indeed, deletion of Notch2 in AKT/Yap-induced tumors switched the phenotype from iCCA to hepatocellular lesions (adenomas or HCC). In contrast, inactivation of Notch1 in hepatocytes did not lead to any significant histomorphological modification of the original lesions (iCCA) in this model, identifying Notch2 as the major determinant of hepatocyte-derived iCCA (69). Obviously, further studies are necessary for a better understanding of the transdifferentiation process in CCA.
Reviewer 2 Report
In my opinion the manuscript is well written and provides a clearly and timely revision of mouse genetic models to be employed for the study of cholangiocarcinoma. My only suggestion is to add a further paragraph about patient-derived xenografts (PDXs) models currently available in this malignancy.
Author Response
Response to reviewers’ comments for Erice et. al “GENETIC MOUSE MODELS AS IN VIVO TOOLS FOR CHOLANGIOCARCINOMA RESEARCH’’ ([Cancers] Manuscript ID: cancers-613344)
We thank the three reviewers for their detailed and helpful comments. We have now addressed the issues raised with significant changes to the manuscript where appropriate. We are confident that the revised version of the manuscript has been significantly improved and that all the comments have been properly addressed.
REVIEWER 2:
In my opinion the manuscript is well written and provides a clearly and timely revision of mouse genetic models to be employed for the study of cholangiocarcinoma.
My only suggestion is to add a further paragraph about patient-derived xenografts (PDXs) models currently available in this malignancy.
We thank the reviewer for bringing up this point. Given spatial constraints and two very recent reviews that have focused intensively on xenografts from CCA cell lines as well as patient-derived tissue xenografts (PDXs) (Cadamuro et al, Clin Res Hepatol Gas. 2018; Loeuillard et al, Biochim Biophys Acta Mol Basis Dis. 2018), the main goal of this review is to provide a comprehensive description of genetic mouse models. Although we do not discuss these approaches in detail, we have now included relevant basic information concerning these models as part of the new Table 3. This table integrates a comparative analysis of the main features of in vivo mouse models, ranging from genetic mouse models to human xenografted models.
Reviewer 3 Report
The review of Erice et. Al is well written; it faces the huge number of mouse models available for Cholanciocarcinoma. They explained clearly pros and cons of each model. I think that a table which summarizes details of each model should help the readers (time of tumor development after injection/creation, microenvironment involvement…).
Authors should linger on the high costs of all the models reported (long time after tumor development, many mice for each experiment).
Moreover, these models are more suitable to study tumor development rather than “co-clinical trial, in particular drug treatment. Nevertheless, being immunocompetent mice, they could be used to test the efficacy if immunotherapies (anti- PD-1, Checkpoint inhibitors). Please, add a paragraph about this topic.
The microenvironment seems to play a key role in tumor development. Which model is the best to study the interaction with microenvironment?
It would be useful to add a comparison of all these models with xenografts and PDXs.
Are the models described all ICC or HCC/ICC? Are there any ECC models?
These models are very important tool to study the tumor arising and progression and authors described them very well.
Author Response
Response to reviewers’ comments for Erice et. al “GENETIC MOUSE MODELS AS IN VIVO TOOLS FOR CHOLANGIOCARCINOMA RESEARCH’’ ([Cancers] Manuscript ID: cancers-613344)
We thank the three reviewers for their detailed and helpful comments. We have now addressed the issues raised with significant changes to the manuscript where appropriate. We are confident that the revised version of the manuscript has been significantly improved and that all the comments have been properly addressed.
REVIEWER 3:
The review of Erice et. Al is well written; it faces the huge number of mouse models available for Cholanciocarcinoma. They explained clearly pros and cons of each model. These models are very important tool to study the tumor arising and progression and authors described them very well.
We are thankful to the reviewer for his/her comment about the original version of the manuscript.
I think that a table which summarizes details of each model should help the readers (time of tumor development after injection/creation, microenvironment involvement…).
We agree with the reviewer that this information is important. We have added additional information to both Table 1 and Table 2 to comply with the reviewer’s request.
Authors should linger on the high costs of all the models reported (long time after tumor development, many mice for each experiment).
We appreciate the reviewer’s comment and have included references to this point both in the text (new Summary section) and in new Table 3, where a comparative analysis of the main features of in vivo mouse models is provided.
Moreover, these models are more suitable to study tumor development rather than “co-clinical trial, in particular drug treatment. Nevertheless, being immunocompetent mice, they could be used to test the efficacy if immunotherapies (anti- PD-1, Checkpoint inhibitors). Please, add a paragraph about this topic.
We acknowledge that our review focuses mostly on the more “technical” and phenotypic aspects of current CCA mouse models, but lacks a thorough description of how murine models have been used in pre-clinical research.
We have now included a brief paragraph in the new Summary section that emphasizes on the potential of genetically flexible model systems to address questions related to current challenges that we frequently encounter when using precision medicine and immune-oncology approaches in CCA patients.
The microenvironment seems to play a key role in tumor development. Which model is the best to study the interaction with microenvironment?
We thank the reviewer for laying out this question, as this is not a trivial issue. In contrast to human cell/tissue-xenografted models, tumors in genetic mouse models arise within the context of an immune-competent, intact microenvironment . While traditional GEM-models are, in part, capable of recapitulating the stepwise tumor development from precursor lesions, which is accompanied by gradual microenvironmental remodelling, the advantage of somatic gene transfer models lies within the genetic flexibility that can, among other benefits, be used to rapidly dissect the influence of specific genetic alterations on the microenvironment. Finally, mouse models have also enhanced our understanding of how the microenvironment itself can shape tumor development, as reflected by the enhanced tumorigenesis in the context of liver injury (e.g. Hspd1 model or combined genetic/chemical models) or distinct tumor “lineage” differentiation depending on the technical approach (e.g. HTVI vs electroporation, ….) We have added a paragraph in which we discuss this issue as part of a new Summary section.
It would be useful to add a comparison of all these models with xenografts and PDXs.
Following the reviewer’s suggestion, we have included this comparison in new Table 3. (Please also see Reviewer 2, Q1 for a more detailed answer).
Are the models described all ICC or HCC/ICC? Are there any ECC models?
We thank the reviewer for the questions and we apologize if this information was not clear in the previous version of the manuscript. Throughout the text we provide several examples of genetic models representative of mostly iCCA, but also models that have the potential to give rise to HCC or HCC/iCCA, in addition. Please see Reviewer 1, Q2 for a more detailed discussion of the different liver cancer histologies that are frequently observed in genetic models. Beyond including a paragraph concerning this issue within the new Summary section, we have also adapted both the text and tables to highlight the histological presentation in each model. To our knowledge, one model representing exclusively extrahepatic CCA exists, and we have included a description of this model within the review (Nakagawa, PNAS 2017). In addition, we now also further highlight the model described by Yamada et al (Hepatology 2015). This model is based on the in vivo transduction of biliary epithelial cells, and we discuss the potential applicability of this system to give rise to extrahepatic CCA upon modification of the injection technique.